# Antioxidant Activity and Photosynthesis Efficiency in *Melissa officinalis* Subjected to Heavy Metals Stress

**DOI:** 10.3390/molecules28062642

**Published:** 2023-03-14

**Authors:** Dorota Adamczyk-Szabela, Ewa Chrześcijańska, Piotr Zielenkiewicz, Wojciech M. Wolf

**Affiliations:** 1Faculty of Chemistry, Institute of General and Ecological Chemistry, Technical University of Lodz, Zeromskiego 116, 90-924 Lodz, Poland; ewa.chrzescijanska@p.lodz.pl (E.C.); wojciech.wolf@p.lodz.pl (W.M.W.); 2Department of Bioinformatics, Institute of Biochemistry and Biophysics Polish Academy of Sciences, Pawińskiego 5a, 02-106 Warsaw, Poland; piotr@ibb.waw.pl

**Keywords:** antioxidant activity, electrochemical oxidation, lemon balm, phenolic compounds, photosynthesis

## Abstract

The aim of this study was to assess influence of cadmium and zinc treatments on antioxidant activity combined with the photosynthesis efficiency in a popular herb lemon balm (*Melissa officinalis* L.). Plants were grown under greenhouse conditions by the pot method. The Mn, Cu, Cd, and Zn contents in soil and plants were measured by HR-CS FAAS. The activity of net photosynthesis, stomatal conductance, transpiration rate, intercellular CO_2_, and index of chlorophyll in leaves were determined for all investigated species. Reduction of the net photosynthesis was observed for cultivations subjected to either Zn or Cd treatments. Phenolic contents were determined by the chemical Folin-Ciocalteu method, while enhanced voltammetric analysis was applied to assess the antioxidant properties of plant extracts. Both of these approaches yielded similar results. Herbal extracts had exceptional antioxidant capacities and were good scavengers of free radicals and reactive oxygen species. Structural similarity of cadmium and zinc facilitated their mutual structural exchange and prompted substantial expansion of phenolics under the mixed Zn and Cd treatments.

## 1. Introduction

Lemon balm (*Melissa officinalis* L.—*M. officinalis*) is a perennial plant of the family Labiatae. It is highly respected for its pronounced antiviral, antibacterial, anticancer, sedative, antidepressant, and antispasmodic activities. Subtle, fresh, and citric taste make *M. officinalis* an interesting kitchen spice, whose high potential is not fully recognized as of yet. It contains polyphenolic compounds, monoterpenoid aldehydes, monoterpene glycosides, triterpenes, sesquiterpenes, tannins, flavonoids, carotenoids, and essential oil [1]. In particular, phenolics exhibit strong antioxidant properties based on their substantial free radical scavenging power [2]. Additionally, their ability to chelate transitional metals has been widely recognized [3]. Due to the high concentration of rosmarinic and caffeic acids, lemon balm may be a more efficient source of antioxidants than other food products, which are present on a market. Notable, a diet rich in products containing phenolic compounds reduces the risk of cancer, strokes, and cardiovascular diseases [4].

The chemical composition of lemon balm and its medical values depend on several factors, such as light intensity, temperature, nutrients, harvest time, and farming practices. Lemon balm grows wildly in Europe, Asia, and Africa. However, only cultivation in controlled conditions yields uniform material of the standard quality [5].

High market demand prompts intensive production, which gives thousands of tons of dried herb [6]. Lemon balm farming is stimulated by the remarkable low essential oil output, which, on average, is 0.1%, while cultivation yields approximately five metric tons of the dried material per hectare [7]. The largest world producers are Egypt, Turkey, the United Kingdom, and the United States [8]. Poland’s production is among the largest in Europe and approaches 1500 tons annually [9]. Substantial parts of crops are used for essential oil production, which is widely used in the pharmaceutical and cosmetics industries.

In a world wherein development is inevitably related to increasing energy consumption and intensive agriculture, heavy metals are again becoming pollutants of primary concern [10]. They enter the soil environment mostly from phosphorus fertilizers, sewage, and industrial sludge irrigation water, as well as traffic transport [11]. Heavy metals are involved in complex interactions with soil components, which are difficult to foresee and to be kept under control.

Zinc and cadmium often coexist in the soil environment. Those metals readily enter plant roots and are further transported to its above ground parts. Their uptakes are competitive and should be investigated together. Initial work on the combined Cd and Zn interactions was published by Adamczyk-Szabela et al. [12]. This contribution tackles heavy metal uptake and transport in relation to the photosynthesis efficiency. Without knowledge of their mutual interactions, zinc and cadmium migrations cannot be understood in a thorough way. Moreover, high levels of cadmium and zinc in soil induce oxidative stress in exposed plants. The latter prompts synthesis of reactive oxygen species [13], whose destructive activity is neutralized by the elevated production of polyphenolic compounds [14,15]. The sole impact on polyphenol production by plants has been initially investigated by Benhabiles et al. [16] and Kumar et. al. [17]. However, they did not assess the photosynthesis yield on the plant metabolism. The latter process is crucial as far as plant development and polyphenol synthesis is concerned.

To the best of our knowledge, the impact of anthropogenic species in soil on the phenolics content in herbs has not been thoroughly investigated as of yet. Notably, there is still uncertainty concerning functions of the phenolic compounds and their response to Zn and Cd in soil [14]. This problem cannot be tackled without the holistic analysis of water transport and photosynthesis pathways in the herbal plant tissues.

This work, for the first time, applies the electrochemical index introduced by Escarpa [18] to evaluate antioxidant activity of phenolics in the popular herb *M. officinalis*. This approach is combined with exhaustive photosynthesis efficiency estimations, as represented by the gas exchange parameters. Obviously results of this investigation are of practical value for extensive herbal cultivation development.

## 2. Results

### 2.1. Plant Cultivation

*M. officinalis* was grown in a greenhouse under controlled conditions on the mineral-organic soil (organic matter content = 12%) with pronounced acidic character (Ph = 6.61). Bioavailable forms of manganese, copper, cadmium, and zinc (Mn = 37 ± 5 µg/g; Cu = 4.81 ± 0.23 µg/g; Cd = 3.48 ± 0.06 µg/g; Zn = 16.7 ± 0.9 µg/g), as well as their respective total forms (Mn = 94 ± 4 µg/g; Cu = 8.15 ± 0.41 µg/g; Cd = 7.52 ± 0.43 µg/g; Zn = 34.6 ± 1.5 µg/g), clearly indicated that soil was not contaminated by those metals.

### 2.2. Photosynthesis Parameters

The photosynthesis parameters, namely, net photosynthesis (P_N_), index of chlorophyll (Chl), transpiration €, stomatal conductance (Gs), and intercellular CO_2_ (Ci) augmented by biomass of plant were determined for plants over all cultivations (Figure 1).

In general, *M. officinalis* treated with either cadmium or zinc demonstrated reduced photosynthetic yield. That decrease was more pronounced for Cd than Zn supplementations, with the former element being the critical factor. Transpiration and stomatal conductance were pursued in a similar way and indicated that either water or CO_2_ diffusion through stomata are closely related to one another. The chlorophyll content was stimulated by zinc and restrained by cadmium supplementations. The only dry biomass increase, as referred to by the control sample, was identified for (300Zn) treatment. The highest intercellular CO_2_ concentrations were determined for (1Cd), (6Cd), and (50Zn) series. Quite remarkably, the first two are characterized by low net photosynthesis. In general, C_i_ measured for all samples was less diverse than the remaining photosynthesis parameters. Tolerance indices (TI) for all treatments are in Appendix A. TI is defined as a ratio of the dry plant weights, which were grown under the particular supplementation and that of control plants. All values were below unity and indicated that significant stress influence was detected over all treatments [16].

### 2.3. Heavy Metals Uptake

Cadmium, zinc, copper, and manganese contents in roots and above-ground parts of *M. officinalis* are in Table 1. Copper, zinc, and cadmium were mostly accumulated in roots, while manganese preferred above-ground parts of the plant. The cadmium supplementations at either 1 µg/g or 6 µg/g of decreased Mn and Zn levels in roots and above-ground parts. Notably, cadmium addition did not affect the copper content substantially. A sole zinc treatment did not induce significant changes to cadmium levels, but it did hamper manganese uptake and its further transport within the plant. Copper behaved in the opposite way. Joint cadmium and zinc supplementations altered substantial variations of manganese and copper levels.

### 2.4. Total Phenolic Compounds

Concentrations of the total phenolic compounds (TPC), as determined by the Folin-Ciocalteu method, are in Figure 2. All supplementations prompted TPC increase, with cadmium being the most critical element. In particular, the highest increases were observed for mixed (6Cd + 50Zn) and (6Cd + 300Zn) treatments.

### 2.5. Data Analysis

All parameters were determined in parallel for five independent samples. Bartlett’s and Hartley’s tests were applied to check the equality of variance (STATISTICA 10 PL package). Normality of the data sets was evaluated using the Shapiro-Wilk test [19,20]. The post hoc Tukey’s HDS test was used to assess the statistically significant differences among particular parameters.

### 2.6. Electrochemical Oxidation of Plant Extracts

Representative CV and DPV voltammograms for lemon balm extracts are shown in Figure 3. Oxidation peaks were mainly observed in three distinct E/V regions for the CV: (I) 0.4–1.2; (II) 1.3–1.6; (III) 1.7–1.9 V and, for the DPV, (I) 0.2–0.8; (II) 1.3–1.6; (III) 1.7 V–1.9 V. On the CV voltammograms determined for (300Zn), (1Cd + 50Zn), (1Cd + 300Zn), (6Cd + 50Zn), (50Zn), (6Cd), (6Cd + 300Zn) extracts, three electrooxidation peaks were clearly visible, while, for the control and (1Cd), only two maxima were recorded. Furthermore, no reduction in CV peaks was observed, which indicates the irreversibility of anodic processes (Figure 3A). The DPV ensures higher resolution than the CV. Moreover, adsorbable compounds are not electroactive in this technique and were not visible on the DPV voltammograms. The latter shows three distinct anodic peaks for (50Zn, 6Cd + 300Zn) extracts and two peaks for all remaining samples (Figure 3B). Their shapes demonstrate that lemon balm extracts are characterized by diverse qualitative (E_p_) and quantitative (I_p_) compositions of compounds, which were oxidized within investigated potential ranges. The antioxidant activities of lemon balm extracts were assessed with electrochemical indices (EI) calculated by Equation (1), where I_p_ is directly related to the antioxidant power, and E_p_ has a thermodynamic nature. Total antioxidant activities, as represented by EIs, are in Table 2. For clarity, they can be arranged in the following descending order:

6Cd + 300Zn > 6Cd + 50Zn > 1Cd + 300Zn > 1Cd + 50Zn > 300Zn > 6Cd > 50Zn > 1Cd > Control.

The corresponding potentials, currents, and partial activities are collected in Appendix A.

## 3. Discussion

Cadmium and zinc supplementations boosted total phenolic compounds levels in the lemon balm. That spectacular increase was scarcely correlated with the photosynthesis yield, as defined by gas exchange parameters. Polyphenols are secondary metabolites with a number of phenol structural units. They are crucial components of food, and their significance for human health can be hardly overestimated. Therefore, studies on environmental factors, which affect polyphenol production by herbs, are of practical significance. They are synthesized through the shikimate/phenylpropanoid avenue. In the standard growing conditions, this mechanism may be responsible for almost 20% of the carbon fixation. Shikimate dehydrogenase (SKDH) and glucose-6-phosphate dehydrogenase (G6PDH) are crucial for the production of those pathway precursors [21]. The latter may be stimulated by cadmium. Its interactions with key biosynthetic enzymes, such as the phenylalanine ammonia lyase (PAL), SKDH, G6PDH, and the cinnamyl alcohol dehydrogenase (CADH) were characterized in Withania somnifera [22]. Additionally, polyphenol oxidase (PPO) is involved in the ROS scavenging, and it enhances the plant’s resistance to abiotic stress, as induced by heavy metals [23]. Structural similarity of cadmium and zinc prompts their mutual replacement within protein binding sites. This effect explains high increase in total phenolics in lemon balm subjected to mixed Zn and Cd supplementations. The latter was confirmed by the enhanced voltammetric analysis, which was used to evaluate the antioxidant properties of plant extracts. Polyphenols are electroactive compounds that can be easily oxidized at inert electrodes, which act as electron donor agents.

The CV results are entirely in line with the total phenolic concentrations, as measured by the chemical Folin-Ciocalteu method. The DPV results are partially consistent with that picture, and exceptions are (6Cd + 50Zn) and (6Cd) samples. This effect may result from the ways the adsorption of active molecules on electrodes is accounted for in both methods [24].

Biomass production is represented by tolerance indices, which are decreased upon either zinc or cadmium supplementation. The lowest TI (0.44) was observed for (1Cd + 300Zn) supplementation. Quite remarkably, for the remaining mixed treatments [(1Cd + 50Zn) (6Cd + 50Zn) (6Cd + 300Zn)], higher values were observed (0.81; 0.58; 0.78, respectively). Plant species, such as *Phaseolus vulgaris* [25] and *Miscanthus* [26] treated with the sole cadmium doses, showed significant decrease in their growth parameters. However, contradictory results on plants treated with cadmium and zinc treatments were also reported. Namely, Cherif et al. [27] studied tomato plants, demonstrating that zinc may be synergistic with cadmium at elevated concentrations. It hampered cadmium uptake and finally limited oxidative stress, as triggered by this element. Either zinc or cadmium can inhibit plant growth by reducing photosynthesis, respiration, and water uptake [28,29]. A similar effect was observed in *M. officinalis* for all supplementations, with (50Zn) being less prone. The production of various forms of reactive oxygen species (ROS) under the influence of inductive oxidative stress due to the presence of heavy metals hampers plant growth. The latter may result mainly from abnormal cell division. In addition, heavy metals are cofactors, which play an important role in plant photosynthesis and respiration [16]. In particular, Kovácik et al. [30] found that the activation of the enzyme phenylalanine ammonia-lyase (PAL) increases the concentration of phenolic metabolites in *Matricaria chamomilla*. Decline in plant photosynthesis, as induced by heavy metals stress, is usually combined with stomatal restrictions [31]. In those circumstances, limiting stomata openings reduces the stream of carbon dioxide entering the leaf cells. This process is, to a large extent, controlled by a zinc metalloenzyme carbonic anhydrase, which catalyses reaction between CO_2_ and H_2_O. On the other hand, increased zinc concentrations may damage plant photosynthetic apparatus, while reduced CO_2_ uptake decreases Rubisco activity. Both of these processes are likely to be responsible for observed P_N_ declines. Heavy metal transport from roots to above-ground parts is promoted by water flows, which are controlled by stomata [32]. In *M. officinalis,* all Cd and Zn supplementations induced substantial G_S_ reductions, and the largest were observed for mixed treatments.

Chlorophyll is a primary pigment involved in plant photosynthesis. Several heavy metals (Cd, Cu, Zn, Pb, and Hg), which are responsible for plant stress, may replace magnesium cations at chlorophyll reactive center and hamper its light-harvesting efficiency. On the other hand, the observed decline of Chl content in lemon balm could follow inhibition of enzymes involved in chlorophyll biosynthesis. Moreover, reduced activity of δ-aminolaevulinic acid dehydratase and protochlorophyllide reductase in plants subjected to heavy metals stress were observed by Mukhopadhyay et al. [33]. Metals such as iron or copper generate free radicals in the Fenton process. However, they cannot be replaced by metal ions, which are not prone to oxidation in the same way as zinc or cadmium. Nonetheless, the latter inhibit activities of antioxidative enzymes, especially glutathione reductase and, consequently, facilitate accumulation of ROS, which further disturbs lipid bilayers of the thylakoid membrane in plant chloroplasts [34]. Especially, Smeets et al. [35] reported that cadmium-induced oxidative stress in *Phaseolus vulgaris,* through the direct interaction with antioxidative defense, disrupted the electron transport chain. Moreover, the following activation of the lipoxygenase, an enzyme which stimulates lipid peroxidation, has been reported in a study after cadmium exposure [36].

## 4. Materials and Methods

All chemicals were of analytical grade (Fluka or Sigma-Aldrich, Steinheim, Germany).

### 4.1. Soil Preparation and Analysis

Soil samples were taken from an agricultural area close to Piotrków Trybunalski, Poland (51°25′29′′N, 19°34′24′′E) in June 2021 following the standard [37]. The air-dried soil was sieved through 2 mm stainless steel sieve. The soil pH was measured in KCl solution (1 mol/l) by the potentiometric method [38]. The gravimetric approach was used to determine the organic matter content [39]. Bioavailable forms of metals were analyzed in 0.5 mol/l HCl extracts. Pseudo-total concentrations of metals were determined in soil samples subjected to microwave decomposition in a mixed solution of 6 mL HNO_3_ (65%) and 2 mL HCl (36%). An Anton Paar Multiwave 3000 arrangement was used. Metal concentrations were determined by the HR-CS FAAS (High-Resolution Continuum Source Flame Atomic Absorption Spectrometry) with the Analytik Jena ContraAA 300 apparatus. Each sample was analyzed five times.

### 4.2. Plant Material

The entire experiment consisted of nine cultivations with five pots of 200 g of soil in each series (forty-five samples in total). The first (control) was not subjected to supplementations. The remaining series were samples to which Cd(NO_3_)_2_ and Zn(NO_3_)_2_ solutions were introduced in order to obtain concentrations of metals in the soil: (1Cd)) 1 µg/g Cd; (6Cd) 6 µg/g Cd; (50Zn) 50 µg/g Zn; (300Zn) 300 µg/g Zn; (1Cd + 50Zn) 1 µg/g Cd and 50 µg/g Zn; (1Cd + 300Zn) 1 µg/g Cd and 300 µg/g Zn; (6Cd + 50Zn) 6 µg/g Cd and 50 µg/g Zn; (6Cd + 300Zn) 6 µg/g Cd and 300 µg/g Zn. Approximately 0.1 g of *M. officinalis* seeds from P.H. Legutko company, Poland were sown in each pot. Herbs were grown in a greenhouse under controlled conditions: temperatures 23 ± 2 and 16 ± 2°C for day and night, respectively; the relative humidity was limited to 70–75%; the photo-synthetic active radiation (PAR) during the 16-h photoperiod was restricted to 400 μmol/m^2^ s. All herbs were grown for three months. Then, plants were cut, and the above-ground parts were separated from the roots, washed, and air-dried in a ventilated room.

### 4.3. Plant Morphology and Gas Exchange Parameters

All measurements were collected five times on plants from each pot. The content of chlorophyll in leaves was measured by Konica Minolta SPAD-502, Japan in the red and the near-infrared regions. The activity of net photosynthesis (P_N_), the stomatal conductance (G_S_), the intercellular concentration of carbon dioxide (Ci), and the transpiration (E) were measured with the gas analyzer TPS-2 (Portable Photosynthesis System, USA).

### 4.4. Determination of Heavy Metals in Lemon Balm

Dried plant samples (0.5 g) were decomposed by the microwave mineralization in a closed system (Anton Paar Multiwave 3000) with concentrated HNO_3_ (6 mL) and HCl (1 mL) acid solutions. Roots and above ground parts were treated separately. The manganese, lead, copper, cadmium, and zinc concentrations were determined by the HR-CS FAAS with the Analytik Jena ContraAA 300 spectrometer. All measurements were repeated five times. The certified references material INCT-MPH-2 (mixture of selected Polish herbs) was chosen to validate analytical results [40] (Appendix A).

### 4.5. Spectroscopic Determination of Total Phenolic Compounds

The total phenolic content was determined by the Folin-Ciocalteu method with gallic acid applied as the standard [4]. Dried and grounded herbs (1 g) were extracted twice with 40 mL of methanol (70%). Those two volumes were mixed, centrifuged at 3000 rpm for 10 min, and finally transferred into plastic bottles. An amount of 0.1 mL of that extract, 6 mL of distilled water, and 0.5 mL of the Folin-Ciocalteu reagent were placed in 10 mL volumetric flask and thoroughly mixed. After 3 min, 1.5 mL of saturated sodium carbonate solution was added and made up to the mark with distilled water. The solution was subsequently placed in a thermostat at 40 °C for 30 min. The absorbance was measured at wavelength of 765 nm on the Specol 11, (Carl Zeiss, Jena instrument, Jena, Germany). Total phenolic concentrations were expressed as equivalent amounts of gallic acid.

### 4.6. Electrochemical Analysis of Polyphenolic Compounds Extracted from Lemon Balm Plants

Plant extracts were prepared by treating 50 mg of dried, grounded above ground parts of plants with 25 mL of non-aqueous 0.1 mol/l (C_4_H_9_)_4_NClO_4_ solution in acetonitrile. All samples were kept at room temperature over four days and regularly stirred. Solutions were thoroughly deoxygenated by purging with purified argon gas (99.999%) for 15 min prior to the electrochemical experiments. Argon blanket was maintained over the solutions to supply an inert atmosphere during voltammetric analyses. The cyclic voltammetry (CV) and differential pulse voltammetry (DPV) were used to assess antioxidant capacity of phenolic compounds in plant extracts. The potentiostat/galvanostat Autolab instrument integrated with the GPES software (EcoChemie, Utrecht, The Netherlands) was applied for all voltammetric measurement. A three-electrode system equipped with the reference electrode, the auxiliary platinum wire electrode, and the platinum strip working electrode with a surface area of 0.5 cm^2^ was used. A potential of the working electrode was determined in respect to the ferricinium/ferrocene reference electrode (Fc+/Fc) couple, as recommended by the IUPAC [41]. Reference electrode was a platinum wire immersed in the solution of ferrocene c = 1·10^−3^ mol/L in 0.1 mol/L (C_4_H_9_)_4_NClO_4_ in acetonitrile and positioned in the glass tube with a tiny hole (diameter w 0.2 mm) at the bottom. This hole secured an electrochemical contact between the electrolyte in a reference electrode area and that in a reactor chamber. Coulometric oxidation was used to obtain an equivalent of the ferrocene ion concentration ferrocinium (Fc+) (cFc = cFc+). Determinations of antioxidants was performed by the CV and the DPV within the potential range from 0 to 2.0 V. All curves were recorded with scan rates ranging from 0.01 to 1 V s^−1^. The modulation amplitude 25 mV and pulse width 50 ms were applied for DPV measurements. All experiments were carried out at room temperature [42]. The electrochemical index (EI) based on major voltammetric parameters, namely, peak potential (E_pa_), and peak current (I_pa_), was applied for evaluation of the antioxidant activities of particular plant extracts [18,43]. Lower E_pa_ corresponds to high electron donor ability, while the higher Ipa indicates high amount of electroactive species. The following equation [44] was applied:(1)$$\mathrm{EI}=\frac{I_{\mathrm{pa}1}}{E_{\mathrm{pa}1}}+\frac{I_{\mathrm{pa}2}}{E_{\mathrm{pa}2}}+\ldots +\frac{I_{\mathrm{pan}}}{E_{\mathrm{pan}}}$$
where I_pan_ and E_pan_ correspond to current and potential values for each anodic peak, observed in the CV and DPV voltammograms.

## 5. Conclusions

Cadmium and zinc supplementations boosted total phenolic compounds levels in the lemon balm. Their concentrations were determined by the chemical Folin-Ciocalteu method, while the enhanced voltametric analysis was used to evaluate the antioxidant properties of plant extracts. Both of these approaches have shown good correlation and indicated that herbal extracts of lemon balm have exceptional antioxidant capacities and are good scavengers of free radicals and reactive oxygen species. Structural similarity of cadmium and zinc facilitated their mutual structural exchange and explained high increase in phenolics under the mixed Zn and Cd treatments. That spectacular increase was not reflected by the photosynthesis yield as defined by gas exchange parameters. The major factor was decreased openings of stomata, which limited the amount of intercellular carbon dioxide and mitigated the photosynthesis rate. Finally, the decline in biomass production, as represented by tolerance indices, decreased upon zinc or cadmium supplementations.

## Figures and Tables

**Figure 1 molecules-28-02642-f001:**
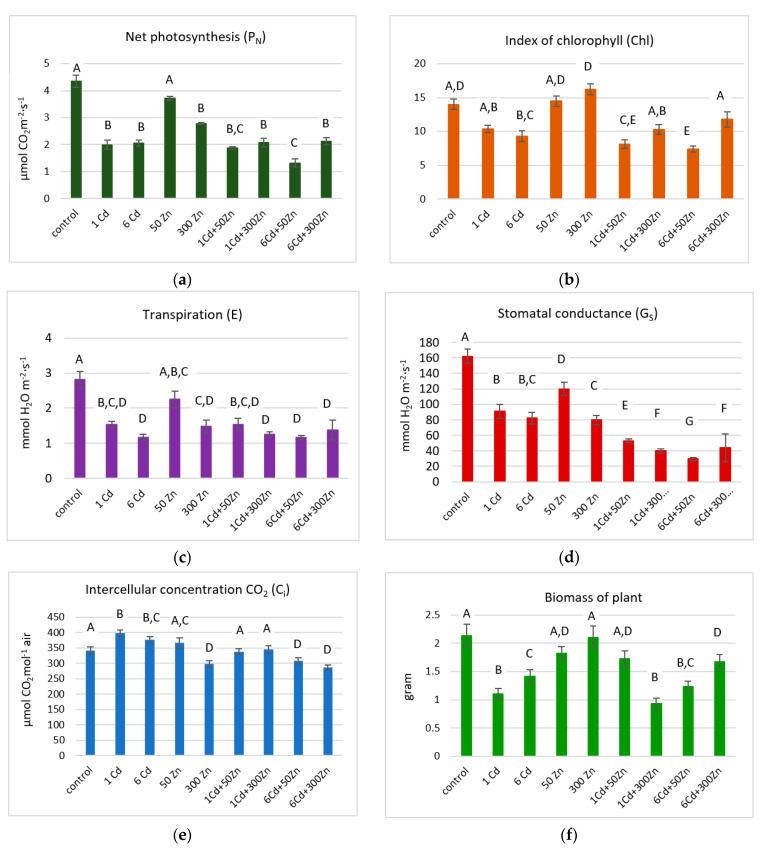
Net photosynthesis (**a**), index of chlorophyll (**b**), transpiration (**c**), stomatal conductance (**d**), intercellular concentration CO_2_ (**e**), and biomass of lemon balm plants (**f**) with relevant standard deviations (*n* = 5). Specific letters illustrate the statistically significant differences as computed with the Tukey’s HSD test (*p* = 0.95). Symbols on the horizontal lines represent Zn and Cd supplementations: (1Cd)-1 µg/g; (6Cd)-6 µg/g; (50Zn)-50 µg/g Zn; (300Zn)-300 µg/g Zn; (1Cd + 50Zn)-1 µg/g Cd and 50 µg/g Zn; (1Cd + 300Zn)-1 µg/g Cd and 300 µg/g Zn; (6Cd + 50Zn)-6 µg/g Cd and 50 µg/g Zn; (6Cd + 300Zn)-6 µg/g Cd and 300 µg/g Zn.

**Figure 2 molecules-28-02642-f002:**
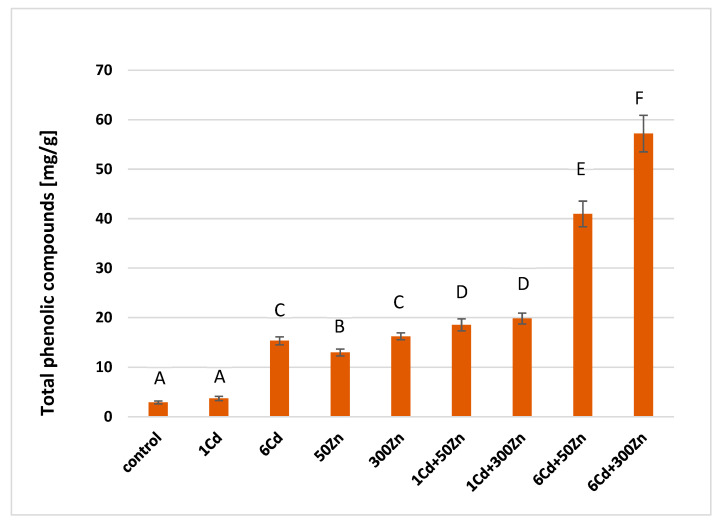
Concentrations of the total phenolic compounds in above ground parts of the lemon balm plants determined for control and all cadmium or zinc treatments with relevant standard deviations (*n* = 5). Specific letters illustrate the statistically significant differences, as computed with Tukey’s HSD test (*p* = 0.95). Symbols on the horizontal lines are the same as described in the caption for Figure 1.

**Figure 3 molecules-28-02642-f003:**
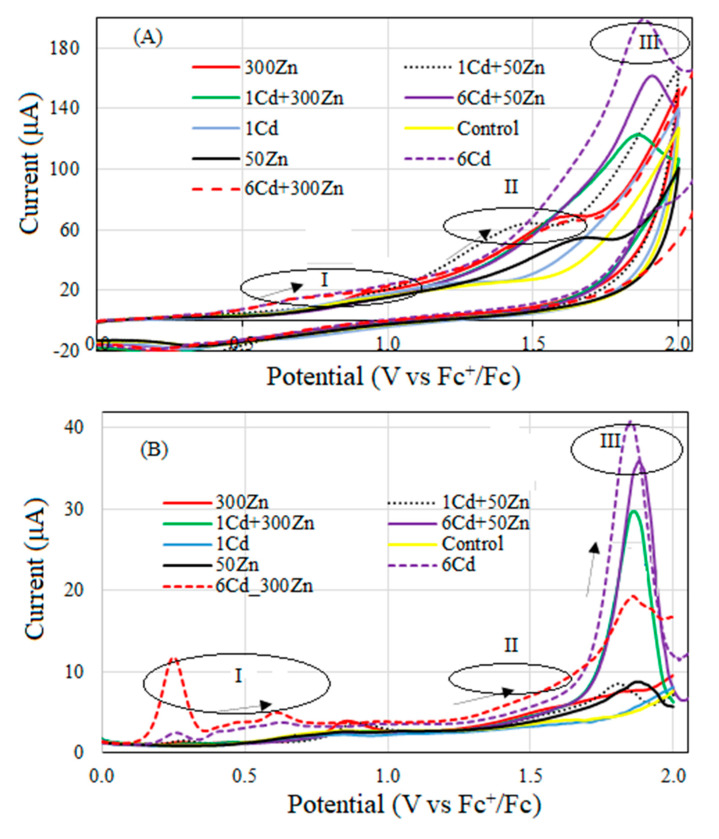
Cyclic (**A**) and differential pulse voltammetry (**B**) voltammograms for lemon balm extracts on Pt electrodes in 0.1 M (C_4_H_9_)_4_NClO_4_. The former were collected with the scan rates v = 0.1 V s^−1^ and v = 0.01 V s^−1^. Modulation amplitude 25 mV and pulse of 50 ms were applied for the latter.

**Table 1 molecules-28-02642-t001:** Copper and manganese contents with relevant standard deviations (*n* = 5) determined in above-ground parts and roots of lemon balm.

Treatments *	Metal Content µg/g							
	Cd		Zn		Cu		Mn	
	Above-Ground Parts	Roots	Above-Ground Parts	Roots	Above-Ground Parts	Roots	Above-Ground Parts	Roots
Control	0.65 ± 0.04	6.82 ± 0.58	29.4 ± 2.3	66.3 ± 5.3	4.87 ± 0.41	8.35 ± 0.72	49.2 ± 4.2	46.7 ± 4.0
1Cd	1.87 ± 0.23	26.3 ± 2.1	19.4 ± 1.8	42.3 ± 2.9	7.06 ± 0.44	9.03 ± 0.85	44.6 ± 3.3	25.1 ± 1.9
6Cd	3.97 ± 0.31	41.5 ± 3.7	14.1 ± 1.7	54.7 ± 5.0	6.18 ± 0.58	7.73 ± 0.63	33.7 ± 2.9	30.4 ± 2.2
50 Zn	0.57 ± 0.04	5.74 ± 0.51	108 ± 7	220 ± 18	6.76 ± 0.53	10.2 ± 0.88	37.9 ± 3.1	25.5 ± 1.3
300Zn	0.52 ± 0.04	7.07 ± 0.43	147 ± 8	981 ± 55	7.18 ± 0.41	12.4 ± 1.1	45.3 ± 3.3	20.8 ± 1.8
1Cd + 50Zn	4.05 ± 0.51	31.3 ± 2.9	51.2 ± 4.4	112 ± 9	7.03 ± 0.58	7.82 ± 0.62	48.6 ± 3.7	16.2 ± 1.3
1Cd + 300Zn	3.05 ± 0.22	38.4 ± 3.0	186 ± 10	427 ± 21	6.39 ± 0.48	9.43 ± 0.83	51.2 ± 4.4	48.3 ± 3.6
6Cd + 50Zn	4.11 ± 0.36	30.8 ± 2.4	65.5 ± 5.8	181 ± 9	7.11 ± 0.55	15.3 ± 1.8	48.4 ± 4.1	24.2 ± 2.0
6Cd + 300Zn	5.12 ± 0.43	39.8 ± 3.4	205 ± 22	655 ± 38	6.15 ± 0.61	12.2 ± 1.2	40.1 ± 3.7	39.3 ± 2.8

***** Symbols used in the Treatments column represent Zn and Cd supplementations: (1Cd)—1 µg/g; (6Cd)—6 µg/g; (50Zn)—50 µg/g Zn; (300Zn)—300 µg/g Zn; (1Cd + 50Zn)—1 µg/g Cd and 50 µg/g Zn; (1Cd + 300Zn)—1 µg/g Cd and 300 µg/g Zn; (6Cd + 50Zn)—6 µg/g Cd and 50 µg/g Zn; (6Cd + 300Zn)—6 µg/g Cd and 300 µg/g Zn.

**Table 2 molecules-28-02642-t002:** Electrochemical indices (EI) determined by CV and DPV methods for lemon balm extracts.

Extract	EI (µA/V) Based on CV	EI (µA/V) Based on DPV
Control	70.5	6.39
1Cd	79.7	6.63
6Cd	104	43.3
50Zn	94.5	11.0
300Zn	119	11.4
1Cd + 50Zn	123	11.5
1Cd + 300Zn	124	18.8
6Cd + 50Zn	153	21.5
6Cd + 300Zn	252	78.0

## Data Availability

The datasets used and analyzed during the current study available from the corresponding author on reasonable request.

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
