# Peer review of "Antioxidant Activity and Photosynthesis Efficiency in Melissa officinalis Subjected to Heavy Metals Stress"

_molecules, 2023, doi:10.3390/molecules28062642_

Round 1

Reviewer 1 Report

The manuscript investigated the effect of Cd and Zn on the antioxidant activity and photosynthesis efficiency of Melissa officinalis, the methods and results are innovative, and the study has practical guidance for the production of Melissa officinalis with high antioxidant properties. However, there are some issues that need to be modified.

These are some comments:

1.    The abstract of the manuscript needs to be rewritten, without explaining why this study was done, and the Abstract contains too much description of Melissa officinalis and too little description of the results of the study.

2.    In lines 83-84, dose total forms of Mn=37±5 µg/g was lower than the bioavailable forms of Mn=94±4 µg/g?

3.    Figure 2 is only the treatments data/control data for the last image of Figure 1, and the data trend is consistent with Figure 1, is it necessary to put it out separately.

4.    The horizontal coordinate 6Cd+5 Zn in Figure 3 should be changed to 6Cd+50 Zn.

5.    Add the explanation of the horizontal coordinates in Figure 1, what 1 cd, 6 cd,,, represent respectively. Label the order of the pictures in Figure 1 with letters, corresponding to the picture serial numbers in the description of the results (lines 87-103).

6.    When Cd and Zn treatments increased the antioxidant level of plants, they also increased their heavy metal content, whether this could cause some adverse effects such as substance extraction and environmental pollution.

7.    Only the total phenolic compounds of the Melissa officinalis was measured and it is recommended to further clarify which phenolics were affected. For the phenolics, the authors interpreted as an indirect response to heavy metal treatment and the effect on plant oxidative stress needs to be supported by data.

8.    The conclusion section contains explanations and predictions, and these are not validated by direct data and are recommended to be modified.

Reviewer 2 Report

Please indicate scientific names in italics (e.g., Melissa officinalis, while "L." is not in italics; Phaseolus vulgaris, etc.)

Acronyms such as HR-CS FAAS must have their meaning indicated in their first appearance.

L46-48 - There is no proper reference for the statistics data indicated.

L69-77 - These sentences have no logical order. First, the authors should indicate the main gap of knowledge to be filled, then explain the main reasons for such a study, and, lately, introduce the objective of their work.

L81-85 - Proper information on the cultivation site is missing in this section to help the reader understand where the experiment was performed, as the M&M section has been placed at the end of the manuscript.

L95 - Figure 1 - proper statistical treatment is missing to verify significant differences between each sample. Please perform appropriate data treatment and indicate the significant letters above each bar graph.

L105 - Figure 2 - Should values on the Y-axis appear with commas or dots?

L116 - Table 1 – Please indicate the meaning of each acronym for your samples in the footnote

L124 – Figure 3 – Same comment as L95; statistical treatment is missing

Round 2

Reviewer 1 Report

1. Delete the last sentence in the abstract.

Reviewer 2 Report

There is no indication on what the "A,A*" means in the caption of Figure 1b. It makes no sense.  

In Figure 2, the letters that indicates the significant differences must follow a proper order regarding the magnitude of each value, not always the alphabetical order, i.e., letters B and C are changed. 
